# Do We Need Adam?
# Surprisingly Strong and Sparse Reinforcement Learning with SGD in LLMs

**Sagnik Mukherjee** [1]  **Lifan Yuan** [1]  **Pavan Jayasinha** [2]  **Dilek Hakkani-Tür** [1]  **Hao Peng** [1]

## Abstract

Reinforcement learning (RL), particularly RL from verifiable reward (RLVR), has become a crucial phase of training large language models (LLMs) and a key focus of current scaling efforts. However, optimization practices in RL largely follow those of next-token-prediction stages (e.g., pretraining and supervised fine-tuning), despite the fundamental differences between RL and these stages emphasized by recent work. One such practice is the use of the AdamW optimizer, which is widely adopted for training large-scale transformers despite its high memory overhead. Our analysis shows that both momentum and adaptive learning rate of AdamW are less influential in RL than in SFT, leading us to hypothesize that RL benefits less from Adam's per-parameter adaptive learning rates and momentum. Confirming our hypothesis, our experiments demonstrate that the substantially more memory-efficient SGD, which is known to perform poorly in supervised learning of large-scale transformers, matches or even outperforms AdamW in RL for LLMs. Remarkably, full fine-tuning with SGD updates fewer than 0.02% of model parameters *without* any sparsity-promoting regularization, more than $1,000\times$ fewer than AdamW. Our analysis offers potential reasons for this update sparsity. Our findings provide fresh insights into the optimization dynamics of RL in LLMs and demonstrate that RL can be substantially more parameter-efficient than previously recognized.

## 1. Introduction

> *"The important thing is not to stop questioning."*
>
> *— Albert Einstein*

[1]University of at Illinois, Urbana-champaign, USA [2]University of Waterloo, Ontario, Canada. Correspondence to: Sagnik Mukherjee <sagnikm3@illinois.edu>.

*Proceedings of the $43^{rd}$ International Conference on Machine Learning*, Seoul, South Korea. PMLR 306, 2026. Copyright 2026 by the author(s).

Reinforcement learning (RL) (Sutton et al., 1998; Ouyang et al., 2022; Ziegler et al., 2020), particularly its verifiable-reward variant (RLVR; Guo et al., 2025; OpenAI et al., 2024), has been a major driver behind the widely recognized success of large language models (LLMs) on complex reasoning tasks (Lightman et al., 2023; Cui et al., 2025; Wang et al., 2025), as well as their alignment with human values and adherence to safety protocols (DeepSeek-AI et al., 2025). Compared to other LLM training paradigms based on next-token prediction (NTP), such as supervised fine-tuning (SFT) and pretraining, RL constitutes a fundamentally different training regime.

Two key differences are particularly relevant. (1) Unlike SFT, online RL samples training data from the most recent version of the policy, causing both the data distribution and the effective optimization landscape to co-evolve with the policy throughout training. (2) RL updates incorporate only $O(1)$ bits of information from the environment per episode, substantially sparser than the $O(\#\text{tokens})$ information in SFT (Schulman and Lab, 2025). These differences have significant impact on the model behaviors as well as the training dynamics. Shenfeld et al. (2025); Chu et al. (2025) demonstrate that RL-trained models generalize better than those trained with SFT, and Chen et al. (2025) attribute RL's better generalization to reduced catastrophic forgetting of on-policy learning. Mukherjee et al. (2025) show that RL fine-tuning updates only about 20% of the parameters which are significantly sparser than those from SFT. Moreover, Zhu et al. (2025) show that RL updates concentrate in off-principal directions of the parameter space, while inducing only minimal spectral drift. Both findings suggest that the effective optimization problem in RLVR is both low-dimensional and geometrically constrained, with learning confined to a subspace of the parameter space.

These differences motivate a closer examination of optimization practices in RL for LLMs, which largely follow those established for NTP stages. Among them, perhaps the most important is the use of the Adam optimizer (Kingma and Ba, 2017), in particular its AdamW variant (Loshchilov and Hutter, 2019; Lambert et al., 2025; OLMo et al., 2025; Grattafiori et al., 2024). Our analysis suggests that both momentum and per-parameter adaptive learning rates, the

*Table 1.* Summary of optimizer update rules and state requirements. $\theta_t$ denotes the parameters at iteration $t$, $g_t$ the gradient, $\eta$ the learning rate, $m_t$ and $v_t$ the first and second moment estimates, $\beta_1, \beta_2$ the moment decay rates, $\varepsilon$ a numerical stability constant. $n$ is the number of trainable model parameters. *For AdamW, for clarity we suppress bias correction and the decoupled weight decay terms.*

| Optimizer | Momentum | Adaptive LR | Final Update | Optim. State |
|---|---|---|---|---|
| SGD | N/A | N/A | $\theta_{t+1} = \theta_t - \eta g_t$ | $O(n)$ |
| SGD + Momentum | $m_t = \mu m_{t-1} + g_t$ | N/A | $\theta_{t+1} = \theta_t - \eta m_t$ | $O(2n)$ |
| RMSProp | N/A | $v_t = \beta_2 v_{t-1} + (1-\beta_2)g_t^2$ | $\theta_{t+1} = \theta_t - \eta \frac{g_t}{\sqrt{v_t}+\varepsilon}$ | $O(2n)$ |
| AdamW | $m_t = \beta_1 m_{t-1} + (1-\beta_1)g_t$ | $v_t = \beta_2 v_{t-1} + (1-\beta_2)g_t^2$ | $\theta_{t+1} = \theta_t - \eta \frac{m_t}{\sqrt{v_t}+\varepsilon}$ | $O(3n)$ |

two key ingredients of AdamW, are less influential in RL than in SFT (§4.2).

These findings lead us to hypothesize that RLVR benefits less from AdamW than SFT, which is supported by our experimental results (§4): ablating from AdamW the first moment (effectively yielding RMSProp; Ruder, 2017), or the second moment (yielding SGD with momentum), or both (yielding SGD) performs on par with or even stronger than AdamW.

Among these findings the most surprising one is the strong performance by SGD in RLVR: SGD has long been considered ill-suited for training large transformers (Pan and Li, 2023; Zhao et al., 2025; Tomihari and Sato, 2025; Zhang et al., 2024; Kunstner et al., 2023) and only works under restrictive settings such as using a very small batch size (Srećković et al., 2025). Our findings suggest that these prior conclusions, usually drawn in supervised learning, may not fully carry over to RLVR for LLMs.

Beyond its strong performance, SGD in RLVR produces highly sparse parameter updates *without* any explicit regularization promoting sparsity (§5). Across three verifiable domains (namely mathematical reasoning, coding and RLVE; Zeng et al., 2025), two model families (Qwen and Llama), and two RL algorithms (PPO and GRPO), SGD updates $0.02\% - 0.46\%$ of model parameters, which is sometimes nearly $500\times$ fewer than AdamW. Our analysis partially attributes SGD's sparser updates to its lack of adaptive learning rates (§5).

Our findings yield several broader insights and implications. First, the pronounced update sparsity with SGD suggests that RL in LLMs can be highly parameter-efficient. The fact that only a small fraction of model parameters are updated offers a mechanistic perspective that complements prior work showing that RL suffers less from reduced catastrophic forgetting (Chu et al., 2025; Chen et al., 2025; Shenfeld et al., 2025) and has a strong dependence on the capabilities of pretrained base models (Gandhi et al., 2025; Yuan et al., 2025; Agarwal et al., 2025). Second, our comparison between AdamW and SGD highlights that optimization decisions depend on the training regime, and that conclusions drawn from SFT may not carry over to RL. From a practi-

cal standpoint, forgoing AdamW's momentum terms yields immediate memory savings. For example, when training the Qwen3-1.7B model, SGD reduces GPU memory usage by 15.7 GB compared to AdamW without losing accuracy (§4.3). Collectively, our findings motivate further investigation into optimization techniques specifically tailored to RL for LLMs, particularly with respect to their potential to reduce forgetting and improve efficiency and scalability. [1]

## 2. Background

In this section, we review policy gradient methods (Sutton et al., 1999) as well as the SGD, AdamW, and RMSProp optimizers, which provide the necessary background for later sections.

**Policy Gradient** To optimize a policy $\pi_\theta$ that maximizes expected rewards, policy gradient methods derive updates by differentiating the objective. For a prompt $\mathbf{x}$, the gradient takes the form:

$$\nabla_\theta J(\theta) = \mathbb{E}_{\mathbf{x}\sim\mathcal{D}, \mathbf{y}\sim\pi_\theta}\left[(R(\mathbf{x},\mathbf{y}) - b)\nabla_\theta \log \pi_\theta(\mathbf{y}|\mathbf{x})\right]$$

(1)

$\theta$ denotes parameters of the policy $\pi_\theta$, and $R$ the return. and $b$ is a baseline for variance reduction, and can be instantiated in various ways: value function estimates, group-averaged returns, or leave-one-out statistics (Shao et al., 2024; Ahmadian et al., 2024). Equation 1 highlights two key attributes of the RL objective: (1) the output trajectory $\mathbf{y}$ is sampled from the evolving policy, creating a non-stationary optimization landscape, and (2) the reward signal $R$ is a rule based reward gained from the environment. In this sense, each episode gains $O(1)$ bits of external information from the environment (Schulman and Lab, 2025).

**SGD, SGD with Momentum, RMSProp, and AdamW** The update rules and state requirements for these optimizers are summarized in Table 1. AdamW maintains two auxiliary states: the first moment (momentum) and the second moment (used for adaptive learning rates). SGD, in contrast, tracks no auxiliary state and has the simplest update rule. Intuitively, RMSProp (Ruder, 2017) and SGD with mo-

---

[1] https://github.com/SagnikMukherjee/sgd_adam_rlvr

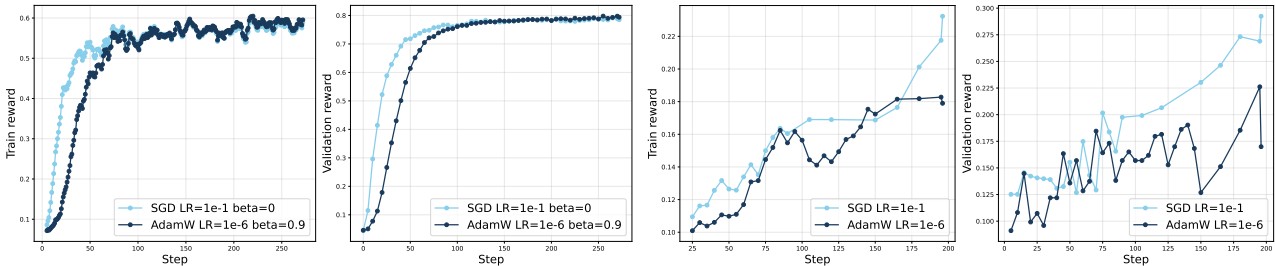

*Figure 1.* Training and validation rewards for SGD and AdamW on math and code tasks. From left to right: training reward for math, validation reward for math, training reward for coding, and validation reward for coding

mentum each retain exactly one of AdamW's components: RMSProp can be viewed as AdamW without momentum (or equivalently, SGD with adaptive learning rates), while SGD with momentum can be viewed as AdamW without adaptive learning rates.[2] Hence, by comparing all four optimizers we can identify which component, if either, is essential for effective RLVR training.

## 3. Do We Need Adam?

As we can see above, the main difference between AdamW and SGD consists of two components: momentum and adaptive learning rate. They are considered beneficial by default, as prior works have extensively demonstrated the superiority of AdamW over SGD. For example, Zhang et al. (2020) argued that adaptive methods provably outperform SGD under heavy-tailed stochastic gradient noise. Pan and Li (2023) attributed AdamW's advantage to favorable directional sharpness properties. Zhao et al. (2025) went further, showing that among common optimizers, SGD uniquely underperforms others for LLM training. A common thread across these explanations is that transformers induce a complex, heterogeneous loss landscape, one with highly varying curvature across parameters, where per-parameter adaptivity becomes essential. However, in light of recent findings that suggest RL has a fundamentally different training dynamics (Mukherjee et al., 2025; Zhu et al., 2025), we revisit this belief. And more specifically we ask:

> **Research Question**
>
> Are adaptive learning rates and momentum needed for RLVR training ?

**Adaptive Learning rate might not be required** AdamW adapts the learning rate for each parameter by normalizing updates using an exponential moving average of squared gradients $\sqrt{v}$.

This increases the effective step size for parameters with historically small gradients and decreasing it for those with

[2]Up to bias correction and weight decay.

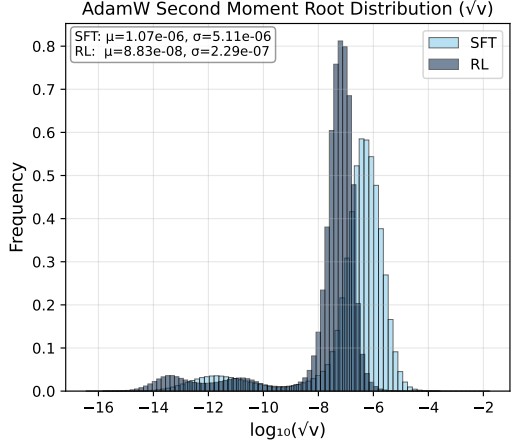

*Figure 2.* Comparison of $\sqrt{v}$ distributions between SFT and RLVR at step 50. RLVR concentrates in a narrower, low-magnitude regime. The standard deviation is $\sim 22\times$ higher in SFT ($\sigma = 5.11 \times 10^{-6}$) than RLVR ($\sigma = 2.29 \times 10^{-7}$).

larger ones. When $\sqrt{v}$ varies substantially across parameters, different parameters experience different effective step sizes. Conversely, if $\sqrt{v}$ is similar across parameters, tracking this auxiliary state confers little benefit beyond single global step size. We first compare the standard deviation in $\sqrt{v}$ between SFT and RLVR training runs on the same model using AdamW (Details in Appendix D). As shown in Figure 2, SFT exhibits approximately $22\times$ higher standard deviation in $\sqrt{v}$ compared to RLVR ($\sigma_{\text{SFT}} = 5.11 \times 10^{-6}$ vs. $\sigma_{\text{RL}} = 2.29 \times 10^{-7}$). This difference suggests that the second-moment, central to AdamW's adaptive learning rates may be far less load-bearing in RLVR than in SFT, motivating us to ablate it later in §4.2.

**Momentum could be counter-productive** Further, we make a crucial observation that RL is fundamentally non-stationary: both the data distribution and the loss landscape evolve throughout training as the policy updates (Sutton et al., 1998). Momentum computes a moving average of past gradients, encoding a memory of previous loss landscapes. However, when data distribution shifts with policy update, the optimization landscape may change substantially

between updates, causing accumulated moment estimates to point in directions misaligned with the current policy gradient. This phenomenon (Bengio et al., 2021) has been shown to hinder optimization in temporal-difference learning and policy gradient methods (Asadi et al., 2023; Ellis et al., 2024; Goldie et al., 2025). Given that RLVR inherits this non-stationarity, the efficacy of momentum-based optimizers in this setting warrants careful investigation.

In order to empirically verify this, we computed cosine similarity between the accumulated momentum $m_{t-1}$ and current step's gradient $g_t$ in SFT and RL in the AdamW optimizer with the code setup discussed in §4. Our analysis (in Appendix E) reveals a striking contrast: in SFT, gradient largely aligns with the accumulated momentum directionally, with a cosine similarity of 0.997 between $g_t$ and $m_{t-1}$; In contrast, in RL, the cosine similarity drops to near-zero ($-0.007$), suggesting substantially weaker directional alignment. These findings provide evidence that RL's non-stationary landscape can make momentum less effective. These two observations lead to our key hypothesis:

> **Hypothesis 1:**
>
> Momentum and adaptive learning rates are less essential in RLVR than in SFT.

## 4. Can SGD Match AdamW in RLVR?

This section empirically evaluates Hypothesis 1. If it holds, we expect SGD, which uses neither momentum nor adaptive learning rates, to achieve similar performance to AdamW (§4.1). We also examine the individual contributions of momentum and adaptive learning rates through additional comparisons with SGD with momentum and RMSProp (§4.2).

**Experimental setup**

- **RL Algorithm:** We experiment with two widely-used RL algorithms: Group Relative Policy Optimization (GRPO; Shao et al., 2024) and Proximal Policy Optimization (PPO; Schulman et al., 2017). Unless otherwise specified, results are reported using GRPO. Experiments in this section are performed with GRPO. PPO results are presented in §6.
- **Domains:** In order to ensure generalizability of our observations, we experiment across three domains: (1) mathematical reasoning, (2) coding, and (3) RL with Adaptive Verifiable Environments (RLVE; Zeng et al., 2025), which contains LeetCode-style synthetic tasks and enables prolonged training.
- **Training datasets:** For math, our training dataset comprises of the NuminaMath-CoT dataset (Jia LI and Polu, 2024) (randomly sampled 35K examples). For coding tasks we use the `code` split (all 25K sam-

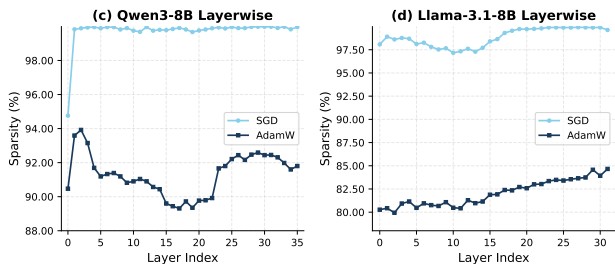

*Figure 3.* SGD updates are distributed across the model rather than concentrated in specific layers. Across all layers, SGD produces significantly sparser updates than AdamW.

ples) of the post-training dataset as used by Cui et al. (2025), where the problems are sourced from APPS (Hendrycks et al., 2021a), CodeContests (Li et al., 2022), TACO (Li et al., 2023), and Codeforces (Codeforces, 2024). Further, for RLVE, we use 260 out of 400 tasks, where each task starts with a difficulty level of 0 and automatically evolves to be more challenging throughout the training.
- **Base models:** We experiment with Qwen3-1.7B, Qwen3-8B (Yang et al., 2025) and Llama-3.1-8B-Instruct (Grattafiori et al., 2024) to study the robustness of our observation across model families and scales.
- **Evaluation:** For math tasks, our evalaute on MATH-500, AMC (Hendrycks et al., 2021b), AIME (Zhang and Math-AI, 2024; 2025), OlympiadBench (He et al., 2024) and GPQA Diamond (Rein et al., 2023). For OlympiadBench we used the OE_MM_MATHS_EN_COMP subset which is comprising of 675 competition level math questions in english. For coding tasks, we evaluate pass@1 and pass@10 on HumanEval (Chen et al., 2021), HumanEval+ (Liu et al., 2023), MBPP (Austin et al., 2021), and MBPP+ (Liu et al., 2023). Pass@K metrics computed with a 0.2 temperature and 10 samples.

Additional details are provided in Appendix A.1. Additionally, SGD requires a much larger learning rate than AdamW, we provide a detailed discussion on that in Appendix B

### 4.1. How Does SGD Fare?

**Math** Table 2 summarizes the results. We evaluate two settings for the maximum rollout length: (1) 3K, matching the training length, and (2) 8K. Across all experiments, SGD closely matches and often outperforms AdamW. Notably, under the 3K rollout setting, SGD consistently outperforms AdamW across all cases.

**Coding** Table 3 presents our results on code generation. We report pass@1 and pass@10. SGD consistently outper-

*Table 2.* Performance comparison of different optimizers on GRPO across model families. SGD achieves comparable or better performance than AdamW. Adaptive learning rates and momentum do *not* consistently improve over SGD.

| Model | Res Len | Optimizer | Math 500 | AIME 24 | AIME 25 | AMC | Olym. | GPQA | Mean |
|-------|---------|-----------|----------|---------|---------|-----|-------|------|------|
| QWEN 3 8B | 3K | AdamW | 89.2 | 50.0 | 43.3 | 60.2 | 51.1 | 39.9 | 55.6 |
| | | SGD | 87.4 | 56.7 | 43.4 | 57.8 | 48.3 | 40.9 | 55.8 |
| | | SGD+Mom | 86.2 | 23.3 | 20.0 | 63.9 | 48.3 | 40.9 | 47.1 |
| | | RMSProp | 89.6 | 60.0 | 40.0 | 63.9 | 49.6 | 41.9 | 57.5 |
| | 8K | AdamW | 93.8 | 63.3 | 66.7 | 74.7 | 60.1 | 58.1 | 69.5 |
| | | SGD | 95.0 | 63.3 | 63.3 | 80.7 | 59.7 | 58.1 | 70.0 |
| | | SGD+Mom | 92.6 | 60.0 | 53.3 | 81.9 | 58.5 | 57.6 | 67.3 |
| | | RMSProp | 94.8 | 83.3 | 63.3 | 72.3 | 60.0 | 53.0 | 71.1 |
| QWEN 3 1.7B | 3K | AdamW | 80.2 | 50.0 | 36.7 | 49.4 | 38.2 | 21.2 | 46.0 |
| | | SGD | 82.6 | 46.7 | 36.7 | 50.6 | 43.1 | 31.8 | 48.6 |
| | | SGD+Mom | 82.8 | 50.0 | 36.7 | 55.4 | 41.2 | 27.3 | 48.9 |
| | | RMSProp | 80.8 | 40.0 | 43.3 | 53.0 | 42.7 | 20.7 | 46.8 |
| | 8K | AdamW | 85.6 | 66.7 | 46.7 | 63.8 | 47.9 | 38.4 | 58.2 |
| | | SGD | 86.2 | 56.7 | 43.3 | 65.1 | 49.2 | 40.4 | 56.8 |
| | | SGD+Mom | 86.2 | 60.0 | 43.3 | 67.5 | 48.6 | 40.4 | 57.7 |
| | | RMSProp | 85.6 | 63.3 | 50.0 | 66.3 | 50.2 | 38.4 | 59.0 |
| LLAMA 3.1 8B | 3K | AdamW | 58.4 | 23.3 | 23.3 | 26.5 | 21.3 | 22.2 | 29.2 |
| | | SGD | 56.2 | 30.0 | 13.3 | 30.1 | 21.5 | 25.3 | 29.4 |
| | | SGD+Mom | 54.8 | 30.0 | 16.7 | 21.4 | 19.6 | 21.7 | 27.4 |
| | | RMSProp | 54.0 | 30.0 | 20.0 | 27.7 | 18.4 | 26.7 | 29.5 |
| | 8K | AdamW | 58.4 | 20.0 | 16.7 | 26.5 | 21.2 | 22.7 | 27.6 |
| | | SGD | 56.2 | 26.7 | 16.7 | 31.3 | 23.0 | 23.7 | 29.6 |
| | | SGD+Mom | 54.4 | 30.0 | 20.0 | 21.7 | 19.6 | 21.7 | 27.9 |
| | | RMSProp | 53.8 | 30.0 | 13.3 | 27.7 | 18.5 | 26.8 | 28.4 |

*Table 3.* Results on coding benchmarks for Qwen3-1.7B trained with GRPO. Training uses a maximum response length of 4K. Evaluations across response lengths of 4K and 8K show that SGD consistently outperforms AdamW on code generation tasks.

| Model | Res Len | Optim. | HumanEval | | HumanEval+ | | MBPP | | MBPP+ | | $\Delta$@1 | $\Delta$@10 |
|-------|---------|--------|-----------|------|------------|------|------|------|-------|------|------|------|
| | | | @1 | @10 | @1 | @10 | @1 | @10 | @1 | @10 | | |
| QWEN 3 1.7B | 4K | AdamW | 44.0 | 54.9 | 37.0 | 47.0 | 39.3 | 64.4 | 46.8 | 72.2 | | |
| | | SGD | **49.3** | **56.1** | **44.1** | **50.6** | **49.6** | **65.2** | **59.0** | **73.0** | +8.7 | +1.6 |
| | 8K | AdamW | 43.7 | 54.9 | 37.0 | 46.3 | 40.9 | 65.2 | 49.0 | 73.5 | | |
| | | SGD | **49.0** | **56.7** | **44.5** | **50.0** | **50.1** | **65.4** | **60.5** | **75.1** | +8.3 | +1.8 |

forms AdamW across all benchmarks and evaluation settings. Under a 4K maximum response length, SGD achieves an average pass@1 improvement of 8.7% over AdamW. Complementing these results, Figure 1 present the learning curves of training and validation rewards while training the Qwen3-1.7B model. They indicate that training with SGD either matches or outperforms AdamW by the end of the training. While these experiments focus on GRPO training for 270 steps, we will soon show in §6 that these findings generalize to extended training duration and PPO.

> **Finding 1**
>
> Despite established wisdom that SGD is ill-suited for transformers (Pan and Li, 2023; Zhao et al., 2025), it performs on par and often outperforms AdamW in training transformers with RLVR.

## 4.2. Do Momentum and Adaptive Learning Rates Help?

As discussed earlier in §2 and Table 1, RMSProp can be intuitively viewed as AdamW ablating momentum, while SGD with momentum can be viewed as AdamW ablating adaptive learning rates. Accordingly, comparisons with them help disentangle the respective contributions of momentum and adaptive learning rates in AdamW, which this section focuses on. Details on the experimental setup for RMSProp and SGD with momentum is detailed in Appendix C .

Table 2 summarizes the results in math reasoning. Comparing SGD + Momentum vs. SGD, we see that momentum hurts the performance in all but one case, with the only exception of Qwen 3 1.7B. This suggests that momentum provides limited benefit in RLVR and may even be counterproductive, consistent with the analysis in §3. RMSProp

*Table 4.* Sparsity and effective rank of parameter updates across different optimizers (A = AdamW, S = vanilla SGD, S+M = SGD with momentum, R = RMSProp) and domains, using GRPO. Across all experiments, SGD-based optimizers induce significantly sparser and lower-rank updates compared to adaptive methods.

| | Math | | | | | | | | Code | | RLVE | |
| | Qwen3-8B | | | | Qwen3-1.7B | | | | Qwen3-1.7B | | Qwen3-1.7B | |
| | A | S | S+M | R | A | S | S+M | R | A | S | A | S |
|---|---|---|---|---|---|---|---|---|---|---|---|---|
| Sparsity | 91.30 | 99.99 | 99.99 | 86.47 | 91.09 | 99.94 | 99.94 | 86.43 | 92.01 | 99.94 | 86.69 | 99.84 |
| Rank | 88.48 | 26.11 | 25.92 | 84.97 | 87.79 | 24.30 | 24.47 | 87.79 | 87.87 | 23.58 | 86.99 | 25.58 |

shows mixed results. It slightly outperforms SGD on Qwen 3 8B and Qwen 3 1.7B at longer response lengths, but underperforms on Llama 3.1 8B. Overall, these results indicate that neither momentum nor adaptive learning rates consistently help in RLVR.

> **Finding 2**
>
> Neither momentum nor adaptive learning rates improves performance.

### 4.3. Memory Footprint of SGD

Memory consumption during the policy update phase of RL is dominated by model weights, activations, and optimizer state. While the first two depend on the model architecture and token count, the optimizer state presents an opportunity for significant memory reduction.

For a model with with $p$ trainable parameters, AdamW requires approximately $12p$ bytes of persistent optimizer state: 4 bytes each for the FP32 master weights, $m$, and $v$. In contrast, SGD requires only $4p$ bytes, as it only maintains the FP32 master weights. Thus SGD reduces memory consumption by $2 \times p \times d_{\text{optim}}$ bytes, where $d_{\text{optim}}$ is the number of bytes per optimizer state element (typically 4 for FP32).

For Qwen3-1.7B, this translates to savings of roughly 13.6 GB in optimizer states. In practice, we observe a 15.7 GB reduction in peak memory usage compared to AdamW on Qwen3-1.7B. The additional savings beyond 13.6 GB reflect the reduced communication buffer overhead incurred by FSDP. This total memory reduction enables training larger models or fitting larger batch sizes within the same hardware constraints.

## 5. SGD Induces Sparse and Low-rank Updates

Now that we have established that SGD achieves competitive performance in RLVR, we next take a closer look at its parameter updates and ask:

> **Research Question**
>
> How do parameter updates induced by SGD compare with those of AdamW in RLVR?

Following Mukherjee et al. (2025), we investigate this question through the lens of *update sparsity*. Let $\theta^0$ and $\theta^1$ denote the model parameters before and after RLVR, respectively. Update sparsity is

$$\text{sparsity}(\theta^0, \theta^1) := 1 - \frac{\|\theta^1 - \theta^0\|_0}{n}$$

$n$ is the number of parameters. The $\ell_0$ norm is computed using a threshold of $10^{-5}$, accounting for numerical precision, taking the `bfloat16` data type into account.[3]

**SGD induces sparser updates than AdamW**   Our results, summarized in Table 4, reveal a striking difference between SGD and AdamW. Across model families and scales, SGD produces orders-of-magnitude sparser parameter updates than AdamW. For example, in the Qwen-3-8B model, AdamW updates approximately 10% of model parameters (corresponding to 90% sparsity), whereas SGD updates only 0.01% of parameters (99.99% sparsity). Similar trends are observed for Qwen-3-1.7B and Llama-3.1-8B, and these observations hold consistently across domains. As shown in Figure **??**, update sparsity under AdamW decreases as training proceeds, while that under SGD barely does.

While SGD with momentum yields update sparsity similar to that of SGD, RMSProp produces sparsity levels comparable to AdamW. These results suggest that the sparsity observed with SGD partly stems from the absence of per-parameter adaptive learning rates. See §7 for a more in-depth discussion.

**Layerwise analysis of update sparsity**   Consistent with the observations of Mukherjee et al. (2025), we find that these sparse updates are *not* concentrated in specific layers or submodules. Figures **??** illustrate the layerwise sparsity, showing that SGD consistently produces significantly sparser updates than AdamW across all layers.

**SGD updates have Low Effective Rank**   Another major difference between SGD and AdamW lies in the effective rank of their parameter update matrices: SGD produces updates with substantially lower rank than AdamW (Table 4).

---

[3]All models are trained with a `bfloat16` precision. PyTorch uses $10^{-5}$ as the default tolerance.

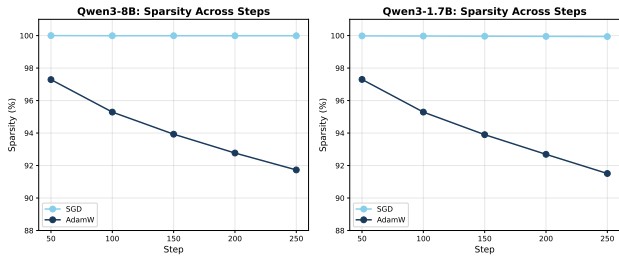

*Figure 4.* Update sparsity of SGD barely decreases as training proceeds. Following plots are from the math experiments

To quantify update rank, we first extract the update matrices corresponding to all two-dimensional weight tensors in the transformer. Then for each update matrix, we perform singular value decomposition (SVD) and compute the number of singular values required to explain 99% of the spectral energy, defined as the sum of squared singular values. This gives us an effective rank per parameter matrix, and we report the mean across all parameter matrices. The results are shown in Table 4. Our observations indicate that models trained with SGD exhibit substantially lower effective rank than their AdamW-trained counterparts. While this difference is less pronounced for the Llama model considered, the overall trend holds consistently across models and settings.

> **Finding 3**
>
> In RLVR, SGD produces highly sparse updates, often modifying only about $0.02\%$ parameters, orders of magnitude fewer than AdamW. The effective rank of SGD updates is also significantly lower than that of AdamW.

## 6. Validation with PPO and Extended Training

To test whether our observations generalize across RL algorithms, we first replace GRPO with PPO and also examine whether they hold under extended training durations. In both settings, we find that SGD achieves performance comparable to AdamW while inducing substantially sparser updates.

*Table 5.* Results on math benchmarks with PPO. The performance across benchmarks is comparable between SGD and AdamW.

| Model | Opt. | Math | AMC | Oly. | GP. | Mean |
|---|---|---|---|---|---|---|
| QWEN 8B | AdamW | 81.8 | 56.6 | 45.6 | 31.8 | 54.0 |
|  | SGD | 80.0 | 55.5 | 44.7 | 46.5 | 56.7 |
| QWEN 1.7B | Adam | 73.2 | 43.4 | 34.2 | 14.6 | 41.4 |
|  | SGD | 73.2 | 37.4 | 30.8 | 26.3 | 41.9 |

**SGD vs. AdamW under PPO** The PPO training setup largely follows that of §4, except for a shorter maximum response length of 1K due to PPO's higher computational cost.

We train both the policy and the critic using the same optimizer. As shown in Table 5, the SGD vs. AdamW comparison under PPO mirrors the trend observed with GRPO, with SGD consistently performing on par with AdamW. The update sparsity observed with GRPO also hold for PPO, where Qwen-3-8B and Qwen-3-1.7B exhibit 99.92% and 99.98% sparse updates (resp.) under SGD and 87.9 and 89.02% sparse updates under AdamW (resp.). Interestingly, the critic in PPO also exhibits substantial sparsity under SGD: for Qwen-3-8B, AdamW updates approximately 61.1% of critic parameters, whereas SGD updates only 8%. This further highlights a qualitative difference between the optimization behavior of SGD and AdamW in RLVR.

**Effects of extended training duration with RLVE** We further evaluate our observations under a setup with substantially more optimization steps to examine whether it exposes potential brittleness of SGD. We choose RLVE because the environment automatically evolves, so models can always train on their capability frontier, preventing the update to stall due to lack of useful supervision signals from data (Zeng et al., 2025). We train with RLVE for 500 steps. Table 6 shows that SGD effectively matches AdamW performance. Notably, even under extended training duration, SGD maintains highly sparse updates: 99.8% of parameters remain unchanged, compared to 86.7% when using AdamW. The competitive performance and substantial sparsity demonstrates the potential of SGD for training over substantially more optimization steps. Further investigation reveals that across training steps the sparsity decays much slower in SGD as compared to AdamW (Figure **??**). Which implies even after prolonged training SGD trained checkpoints will continue to be significantly sparser as compared to AdamW, as also verified in our experiment with the RLVE environment.

> **Finding 4**
>
> The competitive performance and substantial sparsity of SGD generalize to PPO and persist under extended training.

## 7. Why are SGD Updates Sparser?

As noted in prior work (Mukherjee et al., 2025; Zhu et al., 2025), if backpropagation were performed with unlimited numerical precision, the resulting gradients would be very unlikely to be sparse. The update sparsity observed in practice therefore arises from a combination of two factors: (1) many updates having magnitudes close to zero, and (2) these small updates being suppressed by floating-point rounding when applied to the parameters. The latter is an inherent constraint of modern computing hardware and system, it is therefore of particular interest to examine how algorith-

*Table 6.* Results of training on RLVE with GRPO.

| Model | Res Len | Optim. | Math 500 | AIME 24 | AIME 25 | AMC | Olym. | GPQA | Mean | Δ |
|---|---|---|---|---|---|---|---|---|---|---|
| Qwen 3 1.7B | 8K | AdamW | 85.2 | 56.7 | 36.7 | 67.5 | 53.9 | 46.0 | 57.1 | |
| | | SGD | 84.2 | 46.7 | 43.3 | 66.5 | 53.5 | 44.6 | 56.5 | -0.6 |
| | 16K | AdamW | 85.8 | 73.3 | 56.7 | 67.5 | 55.9 | 47.5 | 64.5 | |
| | | SGD | 85.1 | 66.7 | 60.0 | 71.1 | 53.9 | 41.1 | 63.0 | -1.5 |

mic optimization choices in RLVR contribute to the former. Prior work has studied both the inherently small gradients in RL for LLMs (Zhu et al., 2025) and the sparse updates produced by AdamW (Mukherjee et al., 2025). We build on these findings and inquire: why does SGD produce substantially sparser updates than AdamW? We discuss this next.

**The absence of adaptive learning rates** AdamW adapts the learning rate for each parameter by normalizing updates with an exponential moving average of squared gradients, increasing the effective step size for parameters with historically small gradients and decreasing it for those with larger ones. We conjecture that this effectively amplifies updates with small magnitudes that would otherwise be suppressed by floating-point rounding. This conjecture is further supported by the results in Table 4: both SGD and SGD with momentum, which lack adaptive learning rates, produce highly sparse updates; AdamW and RMSProp, both using adaptive learning rates, induce substantially denser updates.

## 8. Related Work

### 8.1. RLVR in LLMs

RLVR has emerged as a key paradigm in the training of LLMs. Removing the noisy reward models, as used in RLHF (Ouyang et al., 2022), it alleviates issues such as reward hacking (Weng, 2024; Amodei et al., 2016; Gao et al., 2022). Further recent work has also established that online RL, as compared to its counterpart supervised finetuning, does not suffer from catastrophic forgetting (Chen et al., 2025; Shenfeld et al., 2025). Owing to these benefits as well as recent algorithmic improvements such as GRPO (Shao et al., 2024), DAPO (Yu et al., 2025) etc, this training paradigm has yielded in significant improvements in expanding reasoning boundaries of LLMs. Recent reports from leading frontier labs report significant effort being spent in RL based post-training of frontier LLMs (xAI, 2025; Yang et al., 2025; Guo et al., 2025; Grattafiori et al., 2024).

### 8.2. Training Dynamics in RLVR

Despite this progress, the training dynamics induced by RLVR in the weight space of LLMs is poorly understood. Some early evidences as discovered by Mukherjee et al.

(2025) show that RL updates are considerably sparser as compared to SFT, where often only 5 to 20% of model weights accumulate any update as part of training. Zhu et al. (2025) argued that this localization happens because online RL causes rotation in the weight space, only altering off-principle eigenvectors. These findings combined paints a picture that RL finetuning happens in a very distinct regime than SFT. Further indicating that borrowing design principles might prove to be suboptimal.

### 8.3. SGD vs AdamW for Training LLMs

Conventional wisdom suggests that, under standard training setups, SGD often underperforms adaptive optimizers such as Adam when training Transformer models. This phenomenon has been well studied, and is often attributed to (i) heavy-tailed distribution of the noise in stochastic gradients (Zhang et al., 2020), (ii) directional sharpness (Pan and Li, 2023) (curvature of the function along the update direction), (iii) Kunstner et al. (2023) provides evidence that this gap in performance is much more visible under a high batch setting, (iv) Zhang et al. (2024) attributed this to the block heterogeneity, i.e. the dramatic difference in the hessian spectrum in the parameter blocks in transformers. Together, these findings provide compelling evidence as to AdamW has become the de facto optimizer for training large (often) transformer based language models.

## 9. Limitations and Future Work

Our investigation shows that SGD can match or outperform AdamW in RLVR. Future work should extend this comparison to a broader class of optimization methods, particularly geometry-aware and structured optimizers such as SOAP (Vyas et al., 2025), Muon-style (Jordan et al., 2024) methods. More broadly, our findings indicate the need for optimizers designed specifically for reinforcement learning in LLMs, rather than directly inheriting choices from pretraining and SFT. RL introduces non-stationary data distributions, sparse reward signals, and distinct update geometry, all of which may call for RL-native optimization principles. Finally, developing training methods that leverage the update sparsity, for example through sparse updates or RL-specific parameter-efficient fine-tuning, is an important direction for improving the scalability and memory efficiency of RL fine-tuning.

## 10. Conclusion

We demonstrate that SGD, long considered ill-suited for training large transformers, matches or outperforms AdamW in RLVR across multiple models and domains and RL algorithms. Neither momentum nor adaptive learning rates consistently improve performance, with momentum often proving detrimental. The observations hold true even under prolonged training. Remarkably, SGD updates fewer than 0.02% of parameters without any explicit sparsity regularization, revealing that effective RL fine-tuning operates in a surprisingly low-dimensional subspace. These findings yield immediate practical benefits—SGD reduces memory usage by up to 15.7 GB compared to AdamW—while highlighting that optimization principles from supervised learning do not directly transfer to RL. We hope this work motivates further investigation into optimization methods tailored to the distinct dynamics of reinforcement learning in LLMs.

## Impact Statement

From a practical standpoint, our work offers immediate memory savings for practitioners training LLMs with reinforcement learning. By eliminating the need for AdamW's momentum buffers, SGD reduces GPU memory footprint, potentially democratizing access to RL-based LLM training for researchers with limited computational resources. The extreme parameter sparsity we observe with SGD also provides mechanistic insights into how RL modifies pretrained models, suggesting that effective reasoning capabilities may emerge from surprisingly localized changes. This understanding could inform future work on efficient fine-tuning methods and help explain why RL-trained models exhibit reduced catastrophic forgetting compared to supervised approaches. We do not foresee specific negative societal consequences arising directly from this work beyond those already associated with LLM training more broadly. Our contributions are primarily methodological and do not introduce new capabilities that would amplify existing risks. If anything, reducing the computational requirements for RL training may help distribute the ability to align and improve LLMs more broadly across the research community.

## Acknowledgements

This work is supported by NSF Grant No. CHE2505932, an Amazon AICE Award, gift funding from AI2, and a grant from Coefficient Giving.

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

# A. Appendix

## A.1. Additional Details of Training Setup

**Models:** For Qwen models, we enabled thinking mode during training, which enables generation of long Chain of thoughts.

**Hyperparameters:** All experiments are implemented using the `verl` framework.[4] Models are trained using four 96 GB NVIDIA GH200 GPUs. Unless otherwise specified, all runs share the following settings: a training batch size of 256, a maximum prompt length of 1,024 tokens and two training epochs. While training models with GRPO we keep the maximum sequence length of 3072 for math (rollouts=4), 4096 for code and 8192 for RLVE. However, in PPO we had to keep the response length to 1024 and 8 rollouts. For experiments with GRPO, we set the KL penalty coefficient to 0.001. All experiments use the KL term as a loss shaping term, and not a reward shaping term. Proximal Policy Optimization (PPO) experiments use Generalized Advantage Estimation (GAE) with a dedicated critic network (learning rate $= 10^{-5}$) and 8 rollout samples. All other training configurations and hyperparameters are held constant, enabling a direct comparison between the two optimizers. For all experiments with AdamW we used a learning rate of $10^{-6}$. Similarly for all experiments with SGD, we used a learning rate of $10^{-1}$. We observed that a much higher learning rate than AdamW is typically required for SGD. Only the PPO experiment with Qwen-3-1.7b required a lower learning rate of $10^{-2}$. However, since AdamW uses a per-parameter adaptive learning rate, the learning rate is not *comparable* across SGD and AdamW.

# B. SGD requires a high learning rate

Our earlier observations indicated that SGD needs a much higher learning rate as compared to AdamW, where the optimal learning for SGD is 0.1 while for AdamW it is $10^{-6}$.

To determine the best operating point for SGD LR, we analyze AdamW's distribution of effective per-parameter learning rates. We compute effective learning rate as $\frac{\eta}{\sqrt{v}+\epsilon}$, where $\eta$ is the nominal learning rate of AdamW, $v$ is the second-moment estimate, and $\epsilon = $ 1e-8. Figure 6 shows this distribution extracted at step 50 of a GRPO code training run. Despite AdamW's nominal learning rate of $10^{-6}$, the effective rates span $10^{-1}$ to $10^{2}$, with the bulk concentrated between $10^{0}$ and $10^{1}$. This explains why SGD requires learning rates $10^{5}$-$10^{6}\times$ larger than AdamW's nominal rate to achieve comparable update magnitudes. This is to be expected since AdamW rescales updates using per-parameter

---

[4] https://github.com/volcengine/verl

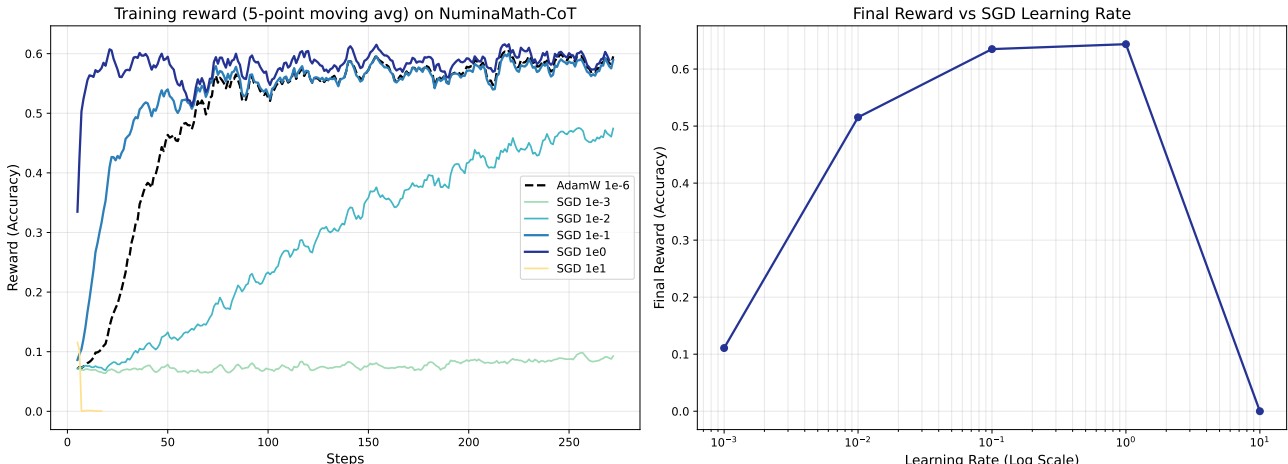

*Figure 5.* **SGD Learning Rate Ablation on Qwen3-8B (NuminaMath- CoT). Left:** Training reward curves over optimization steps. SGD converges to comparable or higher rewards than AdamW, provided the learning rate is sufficiently high. **Right:** Final training reward as a function of learning rate (log scale). SGD requires learning rates orders of magnitude larger than AdamW ($10^{-6}$) to achieve peak performance in the RLVR setting.

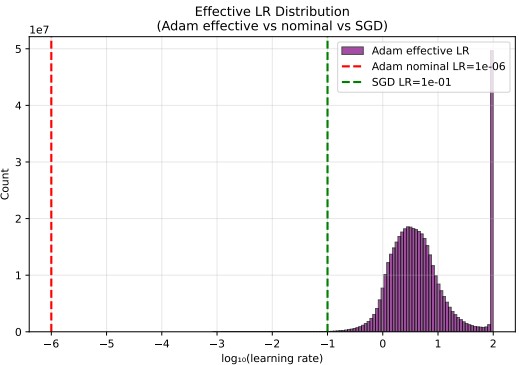

*Figure 6.* **Distribution of AdamW's effective per-parameter learning rates.** We compute the effective learning rate $\eta_{\text{eff}} = \frac{\eta}{\sqrt{v}+\epsilon}$ for each parameter using moment estimates extracted at step 50 of a GRPO code training run (see §4 for setup details). Despite AdamW's nominal learning rate of $10^{-6}$ (red dashed line), the distribution of effective learning rates spans roughly $10^{-1}$ to $10^2$, with the bulk concentrated between $10^0$ and $10^1$. SGD's learning rate of $10^{-1}$ (green dashed line) falls near the lower tail of this distribution, explaining why SGD requires learning rates $10^5$–$10^6\times$ larger than AdamW's nominal rate to achieve comparable update magnitudes.

second-moment estimates–often leading to much larger effective step sizes than suggested by the nominal learning rate.

To empirically confirm the need for high SGD LR, we swept SGD over LR $\in \{10^{-3}, 10^{-2}, 10^{-1}, 1, 10\}$ using the same setup (math tasks) on Qwen3-8B detailed in Section 4. Figure 5 shows that SGD converges to comparable final reward at LR= 0.1 and LR= 1, only crashing at LR= 10. Notably, SGD underperforms at low learning rates (LR=$10^{-2}$), indicating that RL fine-tuning's loss landscape not only tolerates

but benefits from aggressive step sizes.

## C. Experimental Details for ablating momentum and Adaptive Learning Rate

We train Qwen and Llama models with the mentioned optimizers on the math domain, in a training setup same as that described earlier in §4. For SGD+momentum (momentum=0.9), we sweep over three different learning rates ($10^{-1}$, $10^{-2}$, and $10^{-3}$), and report the highest average validation performance. Similarly, for RMSProp, we sweep learning rates ($10^{-5}$ and $10^{-6}$) and report the best mean performance.

## D. Experimental Details for the Adaptive Learning Rate Analysis

SFT was performed on Qwen3-1.7B using the OpenCode-Instruct dataset (Ahmad et al., 2025); RLVR used the same model with the setup described in Section 4. Both training runs used identical AdamW hyperparameters ($\beta_1 = 0.9$, $\beta_2 = 0.999$, lr $= 10^{-6}$, weight decay $= 0.01$, global batch size $= 256$) and were distributed across 4 GPUs using FSDP. At training step 50, we captured the full optimizer state for all 430M trainable parameters on rank 0: gradients $g_t$ immediately before `optimizer.step()`, and AdamW's first moment $m_t$ (exponential moving average of gradients) and second moment $v_t$ (exponential moving average of squared gradients) immediately after. We then compared the distributions of gradient magnitudes $|g|$, first moment magnitudes $|m|$, and second moment roots $\sqrt{v}$ between the two training regimes.

# E. Experimental Details for the Momentum Analysis

To quantify the role of momentum in AdamW's first moment estimator, we profiled optimizer state at training step 50 under two regimes: supervised fine-tuning (SFT) on high-quality code completions from `nvidia/OpenCodeInstruct` (Ahmad et al., 2025) (filtered to test scores $\geq 0.9$) and train with GRPO with KL regularization on code generation tasks. Both experiments used Qwen3-1.7B with identical hyperparameters ($\beta_1 = 0.9$, $\beta_2 = 0.999$, lr $= 10^{-6}$, batch size $= 256$, 4 GPUs with FSDP). We captured gradients $g_t$ before `optimizer.step()` and first moments $m_t$ after, then recovered the momentum buffer $m_{t-1} = (m_t - (1 - \beta_1)g_t)/\beta_1$ to isolate accumulated history from the current gradient. We computed two metrics across 430M parameters: history ratio $r_t = \|m_{t-1}\|/\|g_t\|$ (whether momentum materially affects the update) and directional alignment $\cos\phi_t = \langle m_{t-1}, g_t\rangle/(\|m_{t-1}\| \cdot \|g_t\|)$ (whether momentum reinforces or opposes the gradient).

# F. Learning Rate Sweep for AdamW and SGD

Optimizer comparisons can be sensitive to learning-rate tuning, especially in RLVR where instability appears at large learning rates. We therefore sweep the learning rate for both AdamW and SGD on Qwen3-8B.

Table 7 shows that both optimizers follow a similar stability–performance tradeoff. AdamW improves from 0.192 at $10^{-7}$ to 0.638 at $10^{-5}$, but collapses at $10^{-4}$. SGD improves with larger learning rates and reaches a slightly higher peak reward of 0.644 at 1.0, before collapsing at 10.0. Thus, even after tuning AdamW, SGD matches or slightly outperforms it without momentum or adaptive learning rates.

| Optimizer | Learning Rate | Final Reward |
|-----------|:-------------:|:------------:|
| AdamW | $10^{-7}$ | 0.192 |
| AdamW | $10^{-6}$ | 0.636 |
| AdamW | $10^{-5}$ | 0.638 |
| AdamW | $10^{-4}$ | 0.000 |
| SGD | $10^{-3}$ | 0.111 |
| SGD | $10^{-2}$ | 0.515 |
| SGD | $10^{-1}$ | 0.635 |
| SGD | 1.0 | **0.644** |
| SGD | 10.0 | 0.000 |

*Table 7.* Learning rate sweep for AdamW and SGD on Qwen3-8B.

