# OpenReview forum: "Do We Need Adam? Surprisingly Strong and Sparse Reinforcement Learning with SGD in LLMs"
_ICML.cc/2026/Conference — ICML 2026 spotlight_

### Official Review · Reviewer_ELSp · 2026-03-08

**Soundness:** 2
**Presentation:** 3
**Significance:** 2
**Originality:** 2
**Overall Recommendation:** 3
**Confidence:** 4

**Summary:**

This paper studies the use of the AdamW optimizer in RLVR training. The analysis suggests that both momentum and adaptive learning rates are less influential in RL training. Experimental results show that using SGD or RMSprop can achieve on-par performance while significantly reducing memory consumption, while achieving much more highly sparse updates.

**Compliance With Llm Reviewing Policy:**

Affirmed.

**Final Justification:**

My concerns have been partially resolved. However, it is still unclear why analyzing sparsity is important when comparing SGD and Adam optimizer context

**Key Questions For Authors:**

## Questions
- If SGD exhibits sparser updates than Adam, what can we do or imply from this observation?
- What beta parameters are used in the Adam optimizer? How does SGD compare to Adam with a smaller $\beta_2 (\text{e.g., }\beta_2=\{0.9,0.95\})$ and a smaller $\epsilon=1e-5$, as previously used in traditional RL [1,2]?
- Is such a high learning rate used in SGD (lr=1e-1) still holds in other RLVR training frameworks, such as trl?
- Why did you use such an abnormal temperature $=0.2$ for evaluation settings? How do the performance change when using other standard sampling configurations (e.g., top-p=0.95, temperature=0.6) [3]

## References
---
    [1] Human-level control through deep reinforcement learning. Nature 2015.
    [2] Adam on Local Time: Addressing Nonstationarity in RL with Relative Adam Timesteps. NeurIPS 2024.
    [3] Does Reinforcement Learning Really Incentivize Reasoning Capacity in LLMs Beyond the Base Model? Neurips 2025 Oral.

**Strengths And Weaknesses:**

## Strength
- Strong empirical results spanning different models and training tasks, really demonstrate that SGD can perform on par or even better performance then Adam in RLVR training.
- Useful, practical findings that significantly reduce memory consumption in RLVR training.
## Weaknesses.
- The necessary of using adaptive learning rates is still in-question. The observation that the second-moment variance in RLVR is smaller than in SFT (which only shows at first steps instead of the whole training process) does not necessarily imply that adaptive learning rates are unnecessary. Given that LLMs are typically highly ill-conditioned, which can still make adaptive methods beneficial. An in-depth analysis (theoretical or empirical) of the properties of the Fisher Information Matrix (in Natural Policy Gradient) in LLMs Reasoning is necessary to demonstrate the ineffectiveness of second-momentum methods. Furthermore, RMSprop also demonstrate storng empirical performance compared to SGD while eliminating the need of exhaustive learning rate tuning compared to SGD.
- The sparsity updates analysis in SGD seems irreleveant. It is unclear why do we need to care about sparsity update in SGD versus Adam optimizer.

---

> ### Author Rebuttal · Authors · 2026-03-31
>
> We thank the reviewer for their thoughtful evaluation. We are encouraged that they found our empirical results strong across different models and tasks, and the practical findings on memory reduction useful. We address each concern below.
>
> **Weakness 1. The necessary of using adaptive learning.....**
>
> Thank you for raising these important points. Computing the full Fisher information matrix is computationally prohibitive for LLMs. And a commonly used efficient approximation is its diagonal approximation [1] – essentially the squared gradients. The second moment in AdamW is an EMA over the same squared gradients. Our analysis is thus closely related to the diagonal FIM. We will add a discussion on this connection.
> Our RMSProp/SGD ablation is a more direct comparison, and we observe similar performance across tasks. SGD did not need exhaustive learning rate tuning — in fact, a learning rate of 1e-1 was stable across models and tasks.
>
> [1] Training Neural Networks with Fixed Sparse Masks
>
> **Weakness 2 and Question 1. Relevance of the sparsity update**
>
> The sparsity comparison between SGD and AdamW reveals a structural property of RLVR with both diagnostic and practical value.
> (1) Diagnostic. SGD's extreme sparsity makes visible that far fewer parameters need updating during RL than commonly assumed. AdamW obscures this by amplifying small gradients into non-negligible updates, hiding the low-dimensionality of the training signal, so the underlying low-dimensionality of the RL training signal is not apparent from AdamW runs alone. We view this as a mechanistic insight into RLVR's optimization dynamics in weight space: RLVR operates in a surprisingly low-dimensional subspace.
> (2) Practical. Beyond memory savings of dropping AdamW's moment buffers (Section 4.3), this sparse update suggests the potential for developing training methods that explicitly exploit this structure—for instance, through sparse update schemes or low-rank parameterizations tailored to the RL stage. Given how compute-intensive RL training is today, we believe characterizing this sparsity is a necessary first step toward such methods.
>
> These observations combined (i) opens avenues for immediate efficiency gains and (ii) provides insights into the training dynamics that future work can use. We will expand this motivation in the revision to make the connection between the sparsity analysis and its downstream implications more explicit.
>
> **Question 2: Beta parameters in Adam ? How does SGD compare Adam with smaller beta/epsilon ?**
>
> We used the default beta values in pytorch (0.9, 0.999).
>
>
> We note that the premise may not be entirely accurate — the referenced papers don't suggest smaller beta/epsilon in AdamW; Mnih et al. uses RMSProp (not AdamW), and Ellis et al. note that RL practitioners commonly use a higher ε (1e-5), not smaller..  We ran AdamW with β₂=0.95 and ε=1e-5, and downstream performance was comparable to our default setting:
>
> ||MATH|AMC|OlympiadBench|GPQA|Mean|
> |---|---|---|---|---|---|
> |Adam (low β)|78.6|49.4|39.26|19.7|46.74|
> |Adam (ours)|80.2|49.4|38.22|21.21|47.26|
>
> **Question 3: Is such a high learning rate used in SGD (lr=1e-1) still holds in other RLVR training frameworks, such as trl?**
>
> Thanks for raising this. We acknowledge that exhaustively validating across all training frameworks (verl, slime, trl, etc.) is infeasible within a single paper, and not a common practice in recent RL literature. Our experiments use verl, one of the most widely used RL frameworks.
> More importantly, we argue that the high LR finding is unlikely to be framework-specific. Different training infrastructures differ in distributed execution and memory management, rather than the underlying optimizer math. SGD with lr=1e-1 computes the same gradient step regardless of the infrastructure, assuming equivalent batch sizes and gradient accumulation settings. Therefore, the choice of training framework does not fundamentally alter the optimizer behavior, and we expect the finding to hold broadly across frameworks.
>
> **Question 4: Temperature for evaluation**
>
> Thanks for pointing this out. We initially used a lower temperature as our core metric was pass@1, for which lower temperature is generally recommended [1]. However, we re-evaluated with the configuration - temperature=0.6, top-p=0.95 - as suggested. The results are in the table. Our observations are consistent: SGD is comparable to AdamW, and specifically outperforms AdamW in pass@1.
> |Model|Res Len|Optim.|HumanEval @1|HumanEval @10|HumanEval+ @1|HumanEval+ @10|MBPP @1|MBPP @10|MBPP+ @1|MBPP+ @10|Δ@1|Δ@10|
> |---|---|---|---|---|---|---|---|---|---|---|---|---|
> |Qwen3-1.7B|4K|AdamW|42.7|67.7|37.0|61.0|40.6|66.8|48.3|74.6|||
> |||SGD|47.4|64.6|43.4|61.0|49.1|66.0|59.6|76.5|+7.7|−0.5|
> ||8K|AdamW|44.3|68.3|38.0|61.0|40.8|67.8|49.3|76.2|||
> |||SGD|47.7|65.9|42.1|59.1|49.8|67.4|60.2|76.2|+6.9|−1.2|
>
> [1] Piloting Copilot, Codex, and StarCoder2?

---

> > ### Author Rebuttal · Reviewer_ELSp · 2026-04-03
> >
> > My questions have been partially resolved. From the Adam configurations, it seems the main problems lie in the first momentum. However, I have some additional questions:
> > - The phenomenon that training neural networks operates in a low-dimensional subspace has been extensively observed before [1,2,3,4]. How is it different this time in RLVR training? It seems the sparsity diagnostic does not provide any new insights beyond prior work.
> > -  Can you rigorously explain why such sparsity implies that RLVR training lies in a low-dimensional subspace? The connection is not very clear to me mathematically. (For example, the identity matrix can still have full rank, but most of the entries are zeros).
> > - Also, sparse updates are practically useful. However, my main question is: *why* is it important to analyze *sparsity when comparing Adam and SGD?*
> >
> > The relevance of the sparsity update is still in question.
> >
> > ----
> >     [1] Measuring the Intrinsic Dimension of Objective Landscapes. ICLR 2018.
> >     [2] Intrinsic Dimensionality Explains the Effectiveness of Language Model Fine-Tuning. ACL 2021.
> >     [3] LoRA: Low-Rank Adaptation of Large Language Models. ICLR 2022.
> >     [4] Gradient Descent Happens in a Tiny Subspace.

---

> > > ### Author Response · Authors · 2026-04-03
> > >
> > > Dear Reviewer,
> > > Thanks again for the questions. We would like to clarify the concerns.
> > >
> > > 1. Our findings differ from prior works in two key aspects. Studies such as those cited, along with the lottery ticket hypothesis [1], show that there _exists_ a low-dimensional subspace - optimization on which emulates the full training behavior; such a subspace needs to be _explicitly identified or constructed_. In contrast, Our findings indicate that RLVR, without any regularization or other techniques promoting low dimensions, _naturally leads to sparse updates_. In this sense, we believe our findings are stronger than what prior work entails.
> > >
> > > 2. As the reviewer correctly points out, low rank (as in the cited works) is distinct from sparse (for example AdamW causes sparse but high rank updates). When only a tiny amount of parameters are updated as in the case of SGD, the maximum possible rank is still much smaller than full rank. For example, if only 0.01% of entries in an n×n matrix receive meaningful updates, there aren't enough nonzero entries to fill the diagonal, making the update is low-rank by construction. In contrast, denser updates are sufficient to support full-rank updates.
> > >
> > > 3. Our conjecture (section 7) is that Adam's second moment unnecessarily scales up a large number of updates that would otherwise remain negligible, inflating the effective update footprint; while SGD, without the momentum and adaptive LR is a closer proxy to the intrinsic updates sufficient in RLVR. Comparisons of sparsity of Adam vs SGD provide insights into the learning dynamics of RLVR in terms of how few parameter updates are genuinely needed. And as the reviewer correctly points out, the extreme sparsity of SGD with same or even better accuracy provides actionable practical insights.
> > >
> > >
> > > [1] The Lottery Ticket Hypothesis: Finding Sparse, Trainable Neural Networks

---

### Official Review · Reviewer_QcmC · 2026-03-10

**Soundness:** 3
**Presentation:** 4
**Significance:** 2
**Originality:** 4
**Overall Recommendation:** 4
**Confidence:** 5

**Summary:**

This paper investigates the optimizer preference in reinforcement learning scenarios. By empirically demonstrating that SGD outperforms AdamW in particular RL setups, analytics are built to provide insights and theoretical understandings on update sparsity in this overlooked domain.

**Compliance With Llm Reviewing Policy:**

Affirmed.

**Final Justification:**

I believe this paper presents a wonderful study of an overlooked optimization question in the RL stage. I confirm that all my concerns are resolved during the rebuttal stage and thus recommend acceptance of this paper. I believe this paper can further benefit from advanced theoretical studies and broader empirical studies.

**Key Questions For Authors:**

Q1: I would like to know if the experiments and the proposed conclusion can be performed and validated on at least one MoE model.

Q2: For all numerical results comparing different optimizers in RL, I believe the baseline results (w/o RL) are best to be reported alongside the final performance to demonstrate the substantial improvement of SGD. Table 2 can be enhanced with visual focuses.

Q3: Regarding the momentum analysis in Section.3 and Appendix E, I would like to see a visualization of the misalignment during RL and SFT training, and under different learning rate (schemes) if possible. I would also like to discuss whether the sparsity in updates can be extended to sparsity in gradient signals, which may lead to further ablation studies and techniques.

Q4: In today's post-training practises, SFT cold-starting is commonly used before RL (e.g., DeepSeek-R1). I would like to know whether optimizer states can be inherited from SFT and initialized for RL. The performance comparison under this setting would be very intriguing and extend the scope of this paper.

Q5: The introduction section can be enhanced with a conclusion of findings or contributions to improve clarity.

**Limitations:**

yes

**Strengths And Weaknesses:**

1. Soundness: The claims of this paper, mostly findings from observation and critical insights, are well-grounded by the experiment results, covering typical scenarios in the RL stage of LLMs. Theoretical groundings are comparatively weaker in this paper.
2. Presentation: The paper is well structured with clear writing and beautiful figures and tables. The formulation of the research motivation and findings can be easily identified.
3. Originality: This study has investigated an untouched topic in post-training LLMs and the corresponding optimization preferences.
4. Significance: The discovery of momentum and learning rate being sub-optimal in RL of LLMs can be a novel contribution to the field. Nevertheless, the significance of this study and it contribution to understanding this problem remains limited by the scope of this paper.

---

> ### Author Rebuttal · Authors · 2026-03-31
>
> We sincerely thank the reviewer for their thoughtful comments. We are encouraged that they found the topic original and novel in the post-training literature, the claims well-grounded by experiments, and the presentation clear. We address the raised questions below.
>
> **Q1: I would like to know if the experiments and the proposed conclusion can be performed and validated on at least one MoE model.**
>
> This is a great point and ideally testing our hypothesis on some MoE models would be great. However, MoE models need substantial resources to be trained with RLVR. With limited compute and access to only open sourced resources, this exploration was severely hindered.
> More specifically, In our settings, we tried Qwen1.5-MoE-A2.7B and OLMoE-1B-7B. While the former doesnt fit the available vram in 4 gh200 GPUs, the latter is not a model on which RLVR works. More specifically on the Olmo model, neither AdamW nor SGD optimizer caused the training rewards to improve. We will include a detailed discussion on our limitation section, and this is definitely worth visiting in future work.
> However, our findings on dense models are still quite valuable and brings forward novel insights.
>
>
> **Q2: For all numerical results comparing different optimizers in RL, I believe the baseline results (w/o RL) are best to be reported alongside the final performance to demonstrate the substantial improvement of SGD. Table 2 can be enhanced with visual focuses.**
>
> This is a great suggestion, and in the main paper we will make sure to add the performances of the base models, we are sharing the math and code evaluation performance for the qwen 3 1.7b (w/o RL) model here
>
> > | Model | Res Len | MATH | AMC | OlympiadBench | GPQA |
> > |-------|---------------|------|-----|---------------|------|
> > | Qwen3-1.7B (Base) | 3k | 69.0 | 37.3 | 29.3 | 12.6 |
> > |                   | 8k | 84.6 | 55.4 | 46.2 | 30.8 |
>
> > | Model | Res Len | HumanEval @1 | HumanEval @10 | HumanEval+ @1 | HumanEval+ @10 | MBPP @1 | MBPP @10 | MBPP+ @1 | MBPP+ @10 |
> > |-------|---------|-------------|--------------|--------------|---------------|--------|---------|---------|----------|
> > | Qwen3-1.7B (Base) | 4k | 41.3 | 54.9 | 36.6 | 47.0 | 29.9 | 55.6 | 40.2 | 68.5 |
> > |                   | 8k | 41.3 | 54.9 | 36.3 | 48.8 | 33.3 | 61.0 | 41.3 | 71.7 |
>
>
> **Q3: Regarding the momentum analysis in Section.3 and Appendix E, I would like to see a visualization of the misalignment during RL and SFT training, and under different learning rate (schemes) if possible. I would also like to discuss whether the sparsity in updates can be extended to sparsity in gradient signals, which may lead to further ablation studies and techniques.**
>
> Regarding sparsity of gradients - as noted in [1] (page 8 footnote), the gradients themselves are not sparse, but rather extremely small. Modern computers rely on floating-point arithmetic with limited precision. As a result, updates with very small magnitudes (e.g., absolute values below 10−40) cannot be represented and are effectively discarded.
> Regarding the visualization - This is a great suggestion, and we are running our analyses to share a detailed visualization. Given the short turnaround time of the ICML rebuttal process, as well as the compute requirements in RL experiments we don’t have the results available yet. However, we will include these details in the revision.
>
> [1] "Reinforcement Learning Finetunes Small Subnetworks in Large Language Models"
>
> **Q4: In today's post-training practises, SFT cold-starting is commonly used before RL (e.g., DeepSeek-R1). I would like to know whether optimizer states can be inherited from SFT and initialized for RL. The performance comparison under this setting would be very intriguing and extend the scope of this paper.**
>
> Thank you for this interesting suggestion. We performed this experiment with qwen3-1.7B. And our observations indicate that an SFT cold start on the AdamW optimizer hurts convergence of training rewards significantly. Across steps, the model (with SFT cold-start on adam optimizer states) consistently underperformed the one without cold-start. The exact validation reward on MATH-500 are as follows.
>
> | Step | Baseline Reward | Cold-start Reward |
> |------|----------------|-------------------|
> | 50   | 66.9           | 34.9              |
> | 100  | 68.2           | 44.6              |
> | 150  | 68.5           | 50                |
> | 200  | 70             | 54                |
> | 250  | 71             | 53                |
>
> **Q5: The introduction section can be enhanced with a conclusion of findings or contributions to improve clarity.**
>
> Thank you for this suggestion. We agree that explicitly listing our contributions in the introduction will improve clarity. We initially wove these throughout the introduction, but will add a dedicated contributions summary at the end of the introduction section. We will update the revised manuscript accordingly.

---

> > ### Author Rebuttal · Reviewer_QcmC · 2026-04-01
> >
> > Regarding the base model performance (Q2), I noticed that in some scenarios, AdamW degrades the performance of the base model (e.g., Qwen3-1.7B-Base / 8K / HumanEval+ / @10). Can you elaborate on this observation?
> >
> > My score will be maintained.

---

> > > ### Author Response · Authors · 2026-04-02
> > >
> > > Dear Reviewer,
> > > Thanks for pointing this out. We would like to highlight that this is a one-off situation and not a pattern. Qwen3-1.7B/ 8K / HumanEval+ @10 is the only case out of 16 (res_len, benchmark, k) combinations where AdamW ends up below the base model.
> > >
> > > Notably, SGD does not exhibit this degradation in any setting.

---

### Official Review · Reviewer_JAur · 2026-03-12

**Soundness:** 3
**Presentation:** 3
**Significance:** 3
**Originality:** 4
**Overall Recommendation:** 6
**Confidence:** 3

**Summary:**

The paper considers the important problem of understanding the optimization dynamics in RL for LLMs.

The main insights are --

1. The per parameter learning rate adaptation of AdamW might not be that important for RL than it is for SFT and pre-training stages.

2. Because of this, Vanilla SGD can often perform on-par or even outperform AdamW for RLVR fintetuning.

3. SGD induces much sparser updates than AdamW, and these updates are spread throughout many layers than being concentrated in few layers as in the latter case.

**Compliance With Llm Reviewing Policy:**

Affirmed.

**Final Justification:**

The rebuttal resolved my concerns, so I am happy to maintain the score.

**Key Questions For Authors:**

This might be a question for future work, but it could be interesting to have a discussion on what analogous findings could look like for second order methods like Muon, Soap etc., which are increasingly being used to train LLMs.

**Limitations:**

yes

**Strengths And Weaknesses:**

Strengths.

1. This paper challenges fundamental assumptions about optimization dynamics in RL for LLMs, and opens many interesting research questions for the community to build on.

2. Beyond the conceptual contributions, using SGD instead of AdamW can have much lower memory footprint.

3. The findings can potentially reveal something deeper about RL optimization dynamics even beyond LLMs.

4. The paper is extremely well written and easy to follow.



Weaknesses.

1. While not a weakness as such, it could be interesting to discuss what analogous findings could look like for second order/ geometry aware methods like SOAP, MuON etc., as these are increasingly being used to train LLMs.

---

> ### Author Rebuttal · Authors · 2026-03-31
>
> We sincerely thank the reviewer for their strong endorsement of our work. We are grateful that they found the paper to challenge fundamental assumptions about RL optimization dynamics, to open interesting research questions for the community, and to be extremely well written. We address the raised questions below.
>
> **While not a weakness as such, it could be interesting to discuss what analogous findings could look like for second order/ geometry aware methods like SOAP, MuON etc., as these are increasingly being used to train LLMs.**
>
> Dear Reviewer, this is indeed a great followup research direction.
> Muon presents an interesting case due to its orthogonalization step, and we leave empirical verification to future work. In our work, we focused on commonly used optimizers such as SGD/AdamW/SGD+Momentum, however a detailed controlled study across optimizer choices is definitely worth exploring in future work.
> We will add a detailed discussion on our paper that future work should explore the second order optimizers.

---

> > ### Author Rebuttal · Reviewer_JAur · 2026-04-03
> >
> > Thanks for the discussion about second-order optimizerss

---

### Official Review · Reviewer_VYMX · 2026-03-13

**Soundness:** 2
**Presentation:** 3
**Significance:** 3
**Originality:** 3
**Overall Recommendation:** 5
**Confidence:** 4

**Summary:**

This work claims that the momentum and adaptive learning rate of AdamW may be unnecessary for RLVR, and SGD can match or outperform AdamW across math, coding, and RLVE domains, two model families, and both PPO/GRPO. In addition, the authors show through further analysis that SGD yields highly sparse parameter updates. This has practical implications as SGD can deliver large memory savings without hurting accuracy and stability insights from pretraining may not transfer to the RL fine-tuning setting.

**Compliance With Llm Reviewing Policy:**

Affirmed.

**Final Justification:**

The rebuttal addressed my concerns and I believe this paper should be accepted to the main conference.

**Key Questions For Authors:**

* Given that prior work cited by the authors suggests SGD can benefit from larger learning rates and higher momentum values but also exhibits learning-rate stability tradeoffs, could the authors provide a comparable sensitivity analysis for AdamW as well?
* Could the authors clarify the PPO inner-loop update schedule, including the number of minibatches and epochs per rollout batch? Did they try comparing SGD and AdamW with a few offline steps (eg. 2/4/8)?
* The update sparsity metric used measures cumulative displacement between the initial and final checkpoints, rather than instantaneous per-step sparsity - does the difference between AdamW and SGD hold at the level of individual updates?
* How would the authors expect the sparsity findings extend to other optimizers such as Adafactor, Lion, Muon?

**Limitations:**

Yes

**Strengths And Weaknesses:**

Strengths:
* The paper is generally well-written, with clear motivating analyses by isolating momentum and adaptive learning rate effects of Adam (comparing across SGD/SGD + momentum/RMSProp/AdamW)
* Experiments are generally sound, with authors ablating domains, models and algorithms to support their claims. However, I have concerns about the hyperparameter choices (see Weaknesses and Questions)
* The induced sparsity of updates is an interesting result supporting RLVR operating in a low-dimensional subspace.
* While no new methods are proposed, the work presents new insights about optimizer choice for the RLVR fine-tuning setting.

Weaknesses:
* While certain SGD hyperparameters are swept (namely LR, which the authors note must be a few orders of magnitude higher), the AdamW hyperparameters are kept fixed as default with lr = 1e-6. Momentum is also kept to 0.9 for the SGD+Momentum experiments. Thus the authors show that SGD can match a standard AdamW setup in RLVR; in previous work they've cited (eg. [1]) it is shown that SGD can obtain improved performance at larger learning rates and higher momentum values but suffer from learning rate stability. The authors show the sensitivity of SGD to learning rate in Figure 5, but there isn't a similar analysis for AdamW. It is possible that similar learning rate stability is observed.
* I couldn't see if PPO inner-loop reuse count or minibatch schedule was specified, but I would expect staleness of actor updates to influence stability. This seems like an important hyperparameter to sweep, especially since several standard RL fine-tuning libraries incorporate some asynchronous nature to the actor updates.

[1] Zhao, Rosie, et al. "Deconstructing what makes a good optimizer for language models." arXiv preprint arXiv:2407.07972 (2024).

Minor:
* Bolding best values in Table 2 would make it more readable.
* Line 259: figure -> Figure
* Footnote, Line 328 second column: bflaot16 -> bfloat16

---

> ### Author Rebuttal · Authors · 2026-03-31
>
> We thank the Reviewer for their encouraging remarks that the paper is well-written with clear motivating analyses, that the experiments are sound with thorough ablations across domains, models, and algorithms, and that the sparsity finding is interesting and supports the view that RLVR operates in a low-dimensional subspace. We appreciate the constructive feedback and address each concern below.
>
> **Q1: Learning rate sweep for AdamW.**
>
> We conducted a learning rate sensitivity analysis for both AdamW and SGD with Qwen3-8B, which is an extension of figure 5 with additional ablations on AdamW. Both optimizers show a similar learning rate stability tradeoff, with SGD’s peak slightly higher than AdamW’s.
>
> > | Optimizer | Learning rate | Final reward |
> > |-----------|---------------|--------------|
> > | AdamW     | 1e-7          | 0.192        |
> > | AdamW     | 1e-6          | 0.636        |
> > | AdamW     | 1e-5          | **0.638**    |
> > | AdamW     | 1e-4          | 0.000        |
> > | SGD       | 1e-3          | 0.111        |
> > | SGD       | 1e-2          | 0.515        |
> > | SGD       | 1e-1          | 0.635        |
> > | SGD       | 1e0           | **0.644**    |
> > | SGD       | 10            | 0.000        |
>
> Regarding other hyperparameters in AdamW: an exhaustive grid search over betas and epsilon is computationally prohibitive in our resource-constrained setting. However, we performed a training run with beta_2 = 0.95 and epsilon = 1e-5, which are some other hyperparameter choices people opt with AdamW.
> The downstream performance of the model was comparable with the setting in the paper.
>
>
> >|                | math | amc  | olympiadbench | gpqa  | mean  |
> >|----------------|------|------|---------------|-------|-------|
> >| Adam (low beta)| 78.6 | 49.4 | 39.26         | 19.7  | 46.74 |
> >| Adam (ours)    | 80.2 | 49.4 | 38.22         | 21.21 | 47.26 |
>
> **Q2: Could the authors clarify the PPO inner-loop update schedule, including the number of minibatches and epochs per rollout batch? Did they try comparing SGD and AdamW with a few offline steps (eg. 2/4/8)?**
>
> We thank the reviewer for the question. We would like to clarify that the PPO inner-loop update schedule is as follows. We sample 256 prompts per rollout batch with 4 responses per prompt, yielding 1024 total rollout samples. Each PPO epoch processes these in minibatches of size 256, with a per-GPU micro-batch size of 8 across 4 GPUs, resulting in 8 gradient accumulation steps per minibatch update. We run 1 PPO epoch per rollout batch, which corresponds to 4 gradient steps per rollout batch (1024 / 256 = 4 minibatch updates).
> This 4-step off-policy configuration is consistent with standard RLVR practice.
>
> **Q3: difference between AdamW and SGD at the level of individual updates**
>
> We analysed the first 10 steps of training with Qwen 3 1.7b in SGD vs AdamW. Our observations show that AdamW causes denser updates even at a per-step level. Please find our step level sparsity numbers here - (After entire 10 steps. AdamW sparsity - 97.57, SGD - 99.87). We will add these results in our revised version of the paper.
>
> | Step   | Adam   | SGD    |
> |--------|--------|--------|
> | SFT→1  | 0.9922 | 0.9996 |
> | 1→2    | 0.9928 | 0.9995 |
> | 2→3    | 0.9925 | 0.9995 |
> | 3→4    | 0.9941 | 0.9994 |
> | 4→5    | 0.9929 | 0.9994 |
> | 5→6    | 0.9939 | 0.9994 |
> | 6→7    | 0.9944 | 0.9993 |
> | 7→8    | 0.9947 | 0.9994 |
> | 8→9    | 0.9947 | 0.9993 |
> | 9→10   | 0.9947 | 0.9993 |
>
>
> **Q4: How would the authors expect the sparsity findings extend to other optimizers such as Adafactor, Lion, Muon?**
>
> We conjecture both Lion and Adafactor to exhibit similar update sparsity to AdamW, since both amplify small gradients into non-negligible updates—Adafactor through adaptive per-parameter scaling, and Lion by replacing gradient magnitudes with their sign. Muon presents an interesting case due to its orthogonalization step. In our work, we focused on commonly used optimizers such as SGD/AdamW/SGD+Momentum. We leave empirical validation across these optimizers to future work.
>
> **Bolding best values in Table 2 would make it more readable**
>
> We will make sure to highlight the best values in the table 2 which will help the readability of the table
>
> **Line 259: figure -> Figure, Footnote, Line 328 second column: bflaot16 -> bfloat16**
>
> We will fix these typos in our revision.

---

> > ### Author Rebuttal · Reviewer_VYMX · 2026-04-03
> >
> > I thank the authors for running additional ablations largely addressing my concerns and am happy to adjust my score to 'Accept'. I would be curious to see if in asynchronous setups or longer offline steps (eg. 8 mini batch steps) whether SGD performance hurts; it also seems feasible that momentum could hurt in these setups as well.

---

> > > ### Author Response · Authors · 2026-04-03
> > >
> > > Dear Reviewer,
> > > We appreciate the thorough engagement through the rebuttal process. We are glad we could address the concerns, and your feedback definitely improves the manuscript.
> > >
> > > To your point, we would expect both optimizers to suffer from a large number of off-policy steps. We will add a detailed discussion on this ablation in our revision to test out how SGD and AdamW both behave under different number of off-policy steps.

---

### Decision · Program_Chairs · 2026-04-30

**Decision:**

Accept (spotlight)

**Comment:**

This paper presents a provocative and practically significant study of optimizer dynamics in Reinforcement Learning from Verifiable Rewards (RLVR) for Large Language Models. The reviewers generally agree that the work is well-written and provides valuable insights into the efficiency of SGD compared to the standard AdamW, particularly regarding the high update sparsity observed during RLVR. During the discussion period, the authors successfully addressed several technical concerns regarding hyperparameter sensitivity, PPO scheduling, and SFT cold-starting. While one reviewer remains skeptical about the theoretical implications of update sparsity and the necessity of departing from adaptive methods, the majority of the committee finds the empirical evidence and memory-saving potential compelling. As the Area Chair, I find these results highly interesting and believe they will spark significant discussion within the community, potentially justifying an Oral presentation. Overall, this paper challenges the "AdamW-by-default" paradigm in the RL post-training stage and offers a novel perspective on the low-dimensional nature of RLVR optimization.

However, to ensure the work is robust for large-scale applications, the authors should address the discrepancy between their current learning rate settings and those used in state-of-the-art LLMs; specifically, the final version should incorporate experiments or discussions regarding layer-wise $\mu P$ scaling, i.e., lr should be $\eta \sqrt{\frac{d_{out}}{d_{in} }}$ per module and per head, and the search for a base learning rate $\eta$. Furthermore, the inclusion of Muon as a baseline would significantly strengthen the comparison against modern geometry-aware optimizers.